# How PrEP delivery was integrated into public ART clinics in central Uganda: A qualitative analysis of implementation processes

Monique A. Wyatt[1,2]*, Emily E. Pisarski[1], Alisaati Nalumansi[3], Vicent Kasiita[3], Brenda Kamusiime[3], Grace K. Nalukwago[3], Dorothy Thomas[4], Timothy R. Muwonge[3], Andrew Mujugira[3,4], Renee Heffron[4,5], Norma C. Ware[1,6], for the Partners PrEP Program Study Team[¶]

1 Department of Global Health and Social Medicine, Harvard Medical School, Boston, Massachusetts, United States of America, 2 Harvard Global, Cambridge, Massachusetts, United States of America, 3 Infectious Diseases Institute, Makerere University, Kampala, Uganda, 4 Department of Global Health, University of Washington, Seattle, Washington, United States of America, 5 Department of Medicine, University of Alabama at Birmingham, Birmingham, Alabama, United States of America, 6 Department of Medicine, Brigham and Women's Hospital, Boston, Massachusetts, United States of America

¶ The Partners PrEP Program Study Team is listed in the acknowledgments.
* monique_wyatt@hms.harvard.edu

**Data Availability Statement:** The qualitative data illustrating study findings are presented as participant quotes within the paper. While the

## Abstract

Tailored delivery strategies are important for optimizing the benefit and overall reach of PrEP in sub-Saharan Africa. An integrated approach of delivering time-limited PrEP in combination with ART to serodifferent couples encourages PrEP use in the HIV-negative partner as a bridge to sustained ART use. Although PrEP has been delivered in ART clinics for many years, the processes involved in integrating PrEP into ART services are not well understood. The Partners PrEP Program was a stepped-wedge cluster randomized trial of integrated PrEP and ART delivery for HIV serodifferent couples in 12 public health facilities in central Uganda (Clinicaltrials.gov NCT03586128). Using qualitative data, we identified and characterized key implementation processes that explain how PrEP delivery was integrated into existing ART services in the Partners PrEP Program. In-depth interviews were conducted with a purposefully-selected sub-sample of 83 members of 42 participating serodifferent couples, and with 36 health care providers implementing integrated delivery. High quality training, technical supervision, and teamwork were identified as key processes supporting providers to implement PrEP delivery. Interest in the PrEP program was promoted through the numerous ways health care providers made integrated ART and PrEP meaningful for serodifferent couples, including tailored counseling messages, efforts to build confidence in integrated delivery, and strategies to create demand for PrEP. Couples in the qualitative sample responded positively to providers' efforts to promote the integrated strategy. HIV-negative partners initiated PrEP to preserve their relationships, which inspired their partners living with HIV to recommit to ART adherence. Lack of disclosure among couples and poor retention on PrEP were identified as barriers to implementation of the PrEP program. A greater emphasis on understanding the meaning of PrEP for users and its

interview transcripts themselves have been deidentified, they may still contain sensitive information that could potentially compromise participant confidentiality. Requests to access these data will be considered on a case-by-case basis, and should be made in writing to the International Clinical Research Center at the University of Washington, Seattle, USA (Email: icrc@uw.edu). Other materials relevant for analysis appear in supplementary files.

**Funding:** This work was supported by the National Institute of Mental Health (R01MH110296 to RH). The funders had no role in the study design, data collection and analysis, decision to publish, or preparation of the manuscript.

**Competing interests:** The authors have declared that no competing interests exist.

contribution to implementation promises to strengthen future research on PrEP scale up in sub-Saharan Africa.

## Introduction

Rollout of daily oral pre-exposure prophylaxis (PrEP) for HIV prevention has been gaining momentum in sub-Saharan Africa since the World Health Organization (WHO) issued its first guidelines in 2015 [1, 2]. A number of countries, including Kenya, South Africa, Uganda, and Zambia, have adopted PrEP as part of their national HIV prevention strategies, undertaking ambitious delivery campaigns [3–8]. Since 2017, PrEP has been provided to priority populations in Uganda by the Ministry of Health, with support from the US President's Emergency Fund for AIDS Relief [9].

Tailored delivery strategies are important for optimizing the benefit and overall reach of PrEP. PrEP is a particularly appealing HIV prevention strategy for HIV-serodifferent couples–in which one partner is living with HIV and the other is not. An integrated approach of delivering time-limited PrEP in combination with antiretroviral therapy (ART) encourages PrEP use in the HIV-negative partner as a bridge to sustained ART use. Building on the results of the Partners PrEP Study, a clinical trial of PrEP efficacy [10], the Partners Demonstration Project implemented integrated delivery of ART and PrEP for serodifferent couples in Kenya and Uganda from 2012 to 2016. Results showed high levels of ART and PrEP uptake, and virtually no incident HIV infection [11, 12]. Qualitative data suggested that the "couples-focused orientation" of the integrated strategy, which prioritized the specific needs and preferences of couples, may have contributed to its overall success [13]. The integrated strategy has since been scaled up nationally in public health facilities in Kenya, where evidence of the feasibility of large-scale PrEP delivery for serodifferent couples is beginning to emerge [14, 15].

There is a long history of delivering PrEP in ART clinics, where existing clinical expertise, counseling skills, HIV testing protocols, infrastructure and supply chain management systems can be leveraged [7, 16]. While the advantages of building upon established service models are clear, the processes involved in integrating PrEP into ART services are not well understood. This paper evaluates the implementation of an integrated strategy of ART and PrEP delivery for serodifferent couples in central Uganda. Using qualitative data, we identify and characterize key processes that explain how PrEP was incorporated into existing ART services in public health facilities. The broader goal is to deepen understanding of how evidence-based interventions are successfully assimilated into routine practice to inform future implementation efforts in other contexts.

## Methods

### Study background and design

This qualitative study was nested within the Partners PrEP Program, a stepped-wedge cluster randomized trial of integrated PrEP and ART delivery for 1,381 HIV serodifferent couples aged 18 and above who were sexually active with each other and newly diagnosed as serodifferent (Clinicaltrials.gov NCT03586128). The goal was to evaluate the impact of PrEP use on ART initiation and persistence within 12 public health facilities in Kampala and Wakiso districts, in central Uganda (S1 File). The intervention consisted of three components: (1) clinic-wide trainings on PrEP delivery for ART providers; (2) provision of lamivudine/tenofovir disoproxil fumarate (3TC/TDF) as PrEP; and (3) ongoing technical assistance (TA) to facility-

based staff from the Partners PrEP Program study team. The training protocol was developed from Ugandan national guidelines, and customized for serodifferent couples. Study enrollment took place from June 1, 2018 until December 15, 2020.

Overall, PrEP initiation by HIV-negative partners was high; following the launch of the intervention, PrEP initiation was 81% across all facilities. The proportion of PrEP initiators who obtained a refill of pills declined over the study period (42.4% at month 1, and 10.8% at month 6). Nearly all (99.4%) partners living with HIV initiated ART within 90 days of enrollment. High levels of viral suppression among partners living with HIV were also observed in the periods before (82.1%) and after (76.7%) the launch of the PrEP intervention, suggesting that the program did not have any substantial effect on viral suppression [17].

As part of the PrEP program, additional behavioral research aimed at understanding sexual practices, health, HIV risk, and ART and PrEP use was conducted in a subset of couples who initiated ART and PrEP (N = 149). Participants were followed for up-to-24 months. The study design also involved a qualitative component whose goals included examining the integrated delivery approach, from the perspective of serodifferent couples and implementing health care providers.

## Study setting

The qualitative study was carried out at the Infectious Diseases Institute Kasangati research site of Makerere University, located near Kampala, Uganda.

## Sampling and recruitment

A subset of the 149 couples participating in the behavioral research component were purposefully selected to take part in qualitative interviews (see Fig 1). The sampling scheme sought to: 1) include up-to-50 couples from the 12 facilities participating in the trial; 2) ensure the proportion of interview participants from each site reflected the proportion of behavioral research participants from that site; and 3) reflect the gender distribution of participants in the trial. Members of each couple were contacted separately by phone to determine interest in participating. Of the couples contacted, 8 refused to participate in a qualitative interview.

Health care providers who were actively involved in program implementation and employed at a participating facility were identified by members of the study team. Research assistants (RA) contacted three providers from each site to offer participation in the study, ensuring that a variety of roles were represented. Counselors, nurses, clinicians responsible for prescribing and dispensing PrEP and ART, and peer educators/ expert clients were purposefully sampled for qualitative interviews. Provider participants were not members of the study team.

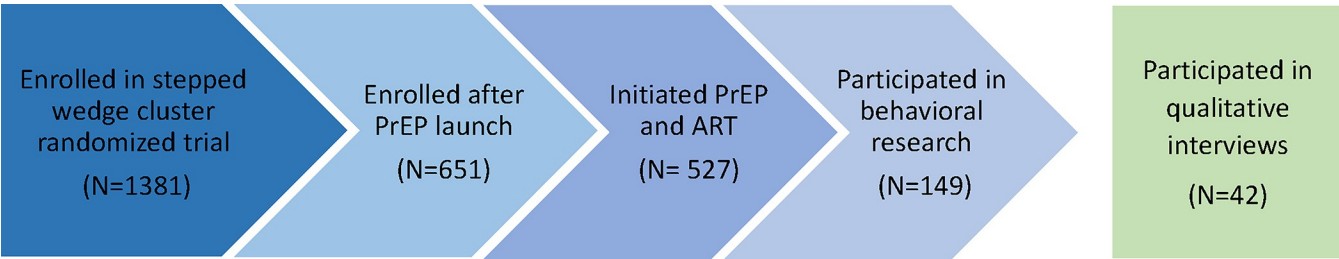

**Fig 1. Serodifferent couples enrollment schema.**

## Data collection

Qualitative data collection took place from September 2019 until July 2021, and consisted of in-depth interviews with implementing providers from each of the 12 facilities (Total: 36 interviews) and individual members of 42 serodifferent couples (Total: 83 interviews). Interviews explored experiences of integrated ART and PrEP delivery. A second interview was done with a subset of 23 couples (Total: 45 interviews) to investigate change over time, and the impact of COVID-19 lockdown restrictions on access to services. The complete qualitative dataset comprised 164 interview transcripts.

The data collection team consisted of two male and three female Ugandan RAs trained in the social sciences and experienced in qualitative methods. They were fluent in the local language and did not know participants personally. Interviews were conducted in English or in Luganda using semi-structured interview guides (S2–S5 Files), and took place in-person in private locations of participants' choosing. During the COVID-19 pandemic, a small number of interviews with members of couples were conducted by telephone (Total: 12). Interviews lasted one hour, on average. They were audio-recorded, with permission, and transcribed directly into English by the interviewer. Interview transcripts were reviewed for quality by author MAW or EEP. Feedback on data quality, including interview technique, content and transcription accuracy was given in weekly teleconference meetings, over email and during in-person supervision visits by senior members of the research team (MAW, NCW, EEP).

## Data analysis

We used multiple approaches to organize and reduce the qualitative interview data for analysis (authors MAW and EEP). For the health care provider interviews, we drafted individual case-based reports summarizing material relevant to integrated ART and PrEP delivery. Illustrative quotes were included in each report. For the data on couples, we identified relevant content to generate code names using open-coding. The names were defined, tested against a sample of transcripts, discussed, refined and compiled into a comprehensive codebook that was then used to guide the coding process (S6 File). Data were coded by author EEP using Dedoose software [18]. We also used a framework approach [19] to reduce the couples data. This involved visually displaying summaries of coded data by interview and pre-designated interview topic in an analytic matrix.

Using the provider case reports, coded couples' data, and data from the matrix, we examined the content inductively to characterize concepts that addressed program implementation and experiences of integrated ART and PrEP delivery. These concepts were used to generate descriptive categories, which involved three distinct steps: 1) developing the category "name"; 2) drafting narrative text elaborating its meaning; and 3) selecting illustrative quotes that explain how the ideas appear in the data. The categories were analyzed inductively to identify their relationships to each other, and were grouped iteratively based on shared content. All concepts relevant to program implementation that were mentioned by participants are represented in the categories; data saturation was achieved.

This combined thematic and content analytic approach revealed three larger conceptual domains describing processes of implementing the integrated strategy of ART and PrEP delivery, described in Results, below. The primary goal of this analysis was to show how health care providers integrated PrEP delivery into ART services, in order to understand key implementation processes. We also explored serodifferent couples' responses to the integrated strategy. The Standards for Reporting Qualitative Research [20] guidelines were used for reporting study findings (S7 File).

## Ethical considerations

Ethical approval to carry out the research was obtained from the Uganda National HIV/AIDS Research Committee, Kampala, Uganda (ARC 194); the Uganda National Council for Science and Technology, Kampala, Uganda (HS 2381); and the University of Washington Human Subjects Division, Seattle, USA (STUDY00000320). Local administrative approval was obtained from the Kampala City Council Authority and Wakiso District.

Members of serodifferent couples provided formal written consent for the qualitative interviews as part of the overall consent process for the behavioral research component of the Partners PrEP Program. Separate written consent was obtained from health care provider participants immediately before each qualitative interview was conducted. Additional information regarding the ethical, cultural, and scientific considerations specific to inclusivity in global research is included as Supporting Information (S1 Checklist).

# Results

## Participant characteristics

A total of 83 individuals in 42 serodifferent couples and 36 health care providers participated in qualitative interviews. One HIV-negative female partner declined to take part in an interview. Median age of participants in the couples' sample was 28 years (interquartile range [IQR]: 24–34 years). Seventy-six percent of members of couples reported being married, and an additional 10% reported they were living with their partner. Sixty-eight percent of women were living with HIV (28 of 41).

For health care providers, the median age was 35 years, and 20% of participants were men. Nearly all providers had pursued education beyond secondary school (92%); forty-seven percent completed at least some university or received a certificate from a tertiary program, 33% completed university, and 11% received a post-graduate degree. The majority were counselors (61%), while 22% were nurses and 3 (8%) were clinicians. The median number of years working at the designated facility was 5 (see Table 1).

**Table 1. Characteristics of qualitative interview participants.**

| Members of serodifferent couples (n = 83) | N (%) or Median (IQR) |
|---|---|
| Age, years | 28 (24–34) |
| Married or living together | 73 (86%) |
| Female | 41 (49%) |
| Living with HIV | 28 (68%) |
| **Health care providers (N = 36)** | |
| Age, years | 35 (31–42) |
| Gender, Female | 29 (81%) |
| Education | |
| Some secondary | 3 (8%) |
| Certificate/ tertiary school | 7 (20%) |
| Some university | 10 (28%) |
| Completed university | 12 (33%) |
| Post-graduate/ Master's | 4 (11%) |
| Profession | |
| Counselors | 22 (61%) |
| Clinicians | 3 (8%) |
| Nurses | 8 (22%) |
| Expert peers | 3 (8%) |
| Time working at facility, years | 5 (4–8) |

## Qualitative results

### Overview of qualitative results

We present three content domains representing the results of the qualitative analysis. Together these domains explain how integrated ART and PrEP delivery was implemented in public health clinics. The domains include: (1) how facility-based providers were supported to integrate PrEP delivery into ART services; (2) how providers promoted interest in integrated delivery for serodifferent couples; and (3) how serodifferent couples responded to integrated ART and PrEP delivery. Discussion of each domain begins with an overview, followed by a detailed presentation of specific concepts comprising each domain.

### How facility-based providers were supported to integrate PrEP delivery into ART services

**Overview.** The experience of integrating PrEP into existing ART services was explored extensively in interviews with implementing providers. While acknowledging a process of adjustment, most spoke about the relative ease with which they and their co-workers adapted to PrEP delivery. Providers pointed to the high-quality training and ongoing supervision and support they received from the PrEP study team as key reasons they were able to successfully carry out the PrEP program. A sense of teamwork among coworkers also strengthened PrEP delivery. Expanded administrative responsibilities felt demanding to some staff, interfering with implementation.

**Building competence for PrEP delivery through high quality training.** Providers largely credited the training that accompanied the launch of the program with helping them to incorporate PrEP delivery into their service package. The training course was appreciated for its scope and "comprehensiveness." Providers were taught practical steps of PrEP provision–i.e., determining eligibility, the significance of adherence, when to discontinue the pills. Importantly, they learned about *how* PrEP works in the body to achieve protection. This helped to dispel initial concerns some providers had about PrEP safety and efficacy, bolstering confidence in their ability to deliver PrEP effectively.

> *It was a new program, a new drug, and most of the health providers did not have knowledge about PrEP. We as health workers were doubting the safety of the drug to our clients and that made it hard for us. However, through the trainings, we acquired knowledge about PrEP and this helped us [trust] the safety of the drug. We are now in position to give out information about PrEP to anyone.*
>
>   -Female counselor, Age 29

**Strengthening motivation through ongoing supervision and technical support.** The Partners PrEP Program team conducted regular visits at each of the 12 facilities to assess the implementation process, review clinical protocols and address challenges. Providers cited the ongoing mentorship and technical support they received during TA visits as highly motivating, and they welcomed feedback on program deliverables (e.g., status of meeting recruitment targets). They especially appreciated the team's responsiveness to facilities' needs, as evidenced by the prompt, uninterrupted supply of PrEP commodities and lab materials (e.g., Hepatitis B and creatinine tests). Providers were encouraged when they were commended for their achievements, and small incentives to celebrate successes made them feel valued. Encouragement reinforced commitments to making the PrEP program work.

*That feedback is encouraging. As you know, everyone works well with motivation. So, whenever you are motivated, whenever you are encouraged, whenever the good that you have done is noticed, at least it gives you hope and motivation to work harder to retain that level that you are at that has made you recognized.*

>-Female clinician, Age 31

**Fostering a flexible approach to implementation through teamwork and collaboration.** Providers attributed their success in integrating PrEP to a number of other factors. Key among these were feeling a sense of "teamwork" with colleagues. Shared roles and responsibilities meant staff functioned in multiple capacities. Focal persons at each facility were responsible for delegating duties to minimize overlap and ensure activities were coordinated and "efficient." Providers described a willingness to step in and help one another when needed. This collaborative approach to service delivery, along with a positive and flexible "attitude," were essential in achieving program goals.

*It is about being determined and if you develop a positive attitude you catch up by also working together with others. Cooperation is important because you cannot identify new couples by yourself; your colleagues need to help.*

>-Female counselor, Age 34

*We work as a team–the laboratory, pharmacy, records, counselors, clinicians. It is basically teamwork and everybody has been involved.*

>-Female expert client, Age 48

**Adding PrEP increases responsibilities for facility staff.** Integration of PrEP delivery did not come without its share of challenges. Initially, some providers were apprehensive that the addition of PrEP would be burdensome and demanding, creating "extra work" they would be unable to manage. A number of providers pointed specifically to the increased documentation and time required for PrEP provision as sources of concern. This was compounded by the need to adhere to different reporting requirements for different PrEP beneficiaries at facilities–i.e., HIV-negative members of serodifferent couples versus other priority populations. Having to juggle new responsibilities without additional staffing support dampened enthusiasm and created resistance among some staff. This tension potentially undermined the success of integrated delivery at some facilities.

*Facility-wise I would say that there was shortage of labor—few staff with a lot of workload. The other thing is that there is a lot of documentation which affects the work, and it also takes time to take this couple through the facility. It consumes a lot of time to work on one couple, yet there is more work to do.*

>-Male counselor, Age 29

## How providers promoted interest in integrated delivery for serodifferent couples

**Overview.** Implementing health care providers felt PrEP filled an important gap in HIV prevention options for serodifferent couples. In interviews, implementers described their investment in and commitment to making integrated delivery appealing to couples. Interest in PrEP was promoted through the ways PrEP and ART use was "packaged" to address the

specific needs and priorities of serodifferent couples. Providers tailored counseling messages, built couples' confidence in the integrated strategy and worked to create demand for PrEP. However, in spite of these efforts, integrated delivery was not an attractive option for all partners, particularly those who were unable or unwilling to disclose their HIV status.

**Affirming the "meaning" of integrated ART and PrEP.** For many couples, the discovery of serodifference was destabilizing, forcing them to grapple with the question of separation. Couples worried that serodifference would make it impossible to remain together, while also keeping the HIV-negative partner safe from HIV. The opportunity to take PrEP in combination with ART through the PrEP program helped to alleviate fears about HIV acquisition, reassuring both partners they could live a long, healthy life together. By restoring hope that relationships could be preserved, integrated delivery fulfilled couples' needs and gave them what they desired most.

> *The program has given me hope to be with my negative partner without worrying of infecting her with HIV. . .. I also have hope of staying with my partner when I am HIV positive while she is HIV negative. My wife is not worried about me infecting her with HIV and we are happy. Personally, I am able to keep my family together because I am sure we would have already separated if we had not joined the program. I am also happy that my wife is given PrEP drugs for prevention. My wife is assured that when she takes PrEP, she cannot contract HIV from me.*
>
> -Male partner living with HIV, Age 31

Implementing providers asserted in interviews that the goal of integrated ART and PrEP delivery was not simply to prevent HIV transmission but also to *prevent separations*. They understood the importance of relationships, and firmly believed PrEP was "the solution" for serodifferent couples "to stay together in love." Consequently, providers promoted PrEP use as both a means of personal protection and an expression of support for partners living with HIV. By appealing to HIV-negative partners' desire to remain in their relationships, providers affirmed the "meaning" of PrEP for couples.

> *The Partners PrEP Program is being implemented because we want to discourage relationship break-ups of discordant couples. It was implemented to encourage people to continue with their relationships irrespective of their different status and also to protect negative people from acquiring HIV, since every positive infection comes from a discordant relationship.*
>
> -Male counselor, Age 31

**Tailoring counseling messages.** Providers at the 12 implementing facilities worked to tailor counseling messages specifically for serodifferent couples. PrEP was presented as an "add on," or an alternative, to condoms, which appealed to couples who preferred "live sex" for enhanced intimacy or who did not wish to use condoms because of conception plans. Adherence to ART and PrEP was emphasized to maintain optimal health for both partners and keep the relationship intact. Partners were encouraged during counseling sessions to be empathetic with one another and were taught practical strategies for providing mutual support. This compassionate counseling approach demonstrated to couples that their *wellbeing* was a priority, which was highly appreciated. They experienced counseling as affirming and responsive to their needs.

> *When you go for your drug refills, the health workers counsel and encourage us to adhere well to our drugs. They also counsel us on how to live with our partners because there is that fear*

*that we get as a result of knowing that your partner is HIV positive. The health workers encourage us to stay with them. [They] counsel us by telling us that it is not the end of the world and encourage us to stay with our partners and not to forsake them. The counseling is within that line of comforting someone.*

> -Female partner taking PrEP, Age 30

**Building confidence in the integrated strategy.** Through counseling and health education, health care providers at implementing facilities worked to make the "evidence" behind integrated ART and PrEP delivery more accessible to couples. Counselors reiterated key messages at each follow up visit, and spoke in clear language that lay people could understand. With concepts that were particularly difficult, like U = U, providers often relied on testimonials from other couples to help communicate how partners remain HIV-negative after PrEP is stopped–i.e., through continued ART use. In interviews, couples demonstrated they understood the counseling messages by explaining, for example, the time it takes for PrEP to reach maximum effectiveness in the body, and the importance of adherence to both ART and PrEP. Many were also able to articulate that partners were protected from HIV acquisition when viral suppression was reached.

*As you know, you can't trust yourself 100%. There are times when I think that maybe one day I could be infected, much as I am on PrEP. But the health workers assured me that I can't [be infected] for as long as her viral load is suppressed. I was told by the health workers that if my wife adheres well to her ART, the virus will reach a time when it is not detected in her blood.*

> -Male partner taking PrEP, Age 30

**Creating demand for PrEP.** To encourage PrEP uptake, clinics developed proactive strategies to identify eligible couples and create demand for PrEP. Some facilities promoted PrEP awareness outside of ART clinics–i.e., in antenatal departments, where couples could test together and learn about their serodifferent status. Others offered PrEP as a "priority service," in which HIV-negative clients were fast-tracked through counseling and laboratory procedures to avoid queues and long waiting times. These efforts were intended to reduce the stigma of being seen in an HIV clinic, so that "healthy" partners would be more willing to consider using PrEP. In giving HIV-negative clients "special attention", facilities showed their commitment to facilitating access to PrEP by improving the service delivery experience.

*We have been giving PrEP more attention, where it includes doing some investigations like Hepatitis B testing among others, because the way you handle a HIV negative partner is quite different from the way you handle a positive person. This is someone who is healthy and you know the queues at the laboratory, so we specifically escort PrEP clients to the laboratory so that they are worked on faster. We get their results and give it to them.*

> -Female expert client, Age 38

**Supporting disclosure within couples.** Counseling for individuals testing positive for HIV in Ugandan public health clinics includes strategies for facilitating disclosure. However, providers at implementing facilities cited lack of disclosure as the primary reason integrated ART and PrEP delivery was not suitable for some serodifferent couples. Some clients living with HIV preferred to avoid relationship conflicts associated with disclosure. Others, especially women, feared abandonment by their partners and the ensuing loss of emotional and/or financial support. To offset these challenges, facilities relied on strategies such as assisted partner

notification (APN) to encourage clients to accept support for the disclosure process. Even with APN counseling, disclosure of HIV status was often not possible, making it difficult for implementers to generate interest in integrated delivery. Without disclosure, at-risk partners who were otherwise eligible for PrEP were unable to benefit from integrated services.

*We get most of our clients through APN. If there is no disclosure, there is no way you can give the client PrEP. . .. We have so many clients who are at risk of getting HIV and yet we cannot disclose to them that their partners are HIV positive since we have to maintain confidentiality. We give these clients condoms, treat them for STIs and yet PrEP would have been the best for them.*

-Female counselor, Age 29

## How serodifferent couples responded to integrated ART and PrEP delivery

**Overview.**   Overall, serodifferent couples at implementing facilities responded positively to the offer of integrated ART and PrEP delivery. In the qualitative sample specifically, individuals living with HIV were motivated to start ART to promote good health, while PrEP initiation was described as a decision to affirm the relationship with a serodifferent partner. PrEP use, in turn, spurred recommitments from spouses living with HIV to adhere to ART to ensure their partners remained free of HIV. Health care providers observed that many individuals at facilities resisted initiating or remaining on PrEP, potentially undermining implementation efforts.

**Initiating PrEP as a sign of solidarity.**   Integrated ART and PrEP delivery was appealing to couples because it eliminated the immediate threat of HIV acquisition. Despite concerns about taking a lifelong, daily medication, most qualitative interviewees living with HIV accepted ART readily. They reported being encouraged by counseling, which reinforced the importance of ART for overall health and longevity. For many HIV-negative partners, the initial decision to accept PrEP signified a commitment to remain in the serodifferent relationship. PrEP use demonstrated solidarity and "oneness" with spouses living with HIV, along with reassurance they would not be abandoned.

*I wanted to make my partner happy so that she may feel there is someone caring for her. You know when I accepted PrEP, she felt that I will not leave her even if she is infected. Taking PrEP indicated to her that we are one and we are still together in a relationship, other than what it would have been if I declined PrEP. She would definitely think that I am leaving her if I do not want to take the medicine that can protect me from getting infected by her.*

-Male partner taking PrEP, Age 24

For HIV-negative partners, taking antiretrovirals alongside partners fostered a sense of mutual understanding and shared experience from gaining firsthand knowledge of what it was like to take a daily medication. This evidence of caring and support had an equalizing effect on the relationship, bringing couples closer together.

*[PrEP] gives them (the partner living with HIV) support. In most cases, when you are taking [pills] and I am also taking, do you know how you feel? There is that comfort that you get that you are not alone. This person understands what I am going through, rather than just sitting in the counselling room with me and telling me, "Sweetheart, you have to take your pills." But now you are with me and you know what it means to take a pill daily. It brings them together and that is one thing that I realized.*

-Female nurse, Age 44

**Recommitting to ART adherence.** This sense of togetherness countered feelings of isolation, and made spouses living with HIV feel appreciated and valued. They were inspired by their partners' PrEP use, and reciprocated by recommitting to ART. In particular, interviewees living with HIV felt a sense of responsibility to their partners, who they understood were taking medication *because of them*. This motivated ART adherence, as they recognized they also had a role to play in ensuring their partners remained healthy. In this way, integrated ART and PrEP delivery encouraged reciprocity within serodifferent relationships, promoting the overall health of both partners.

*Offering PrEP to a negative partner empowers the positive partner to adhere well to the treatment since PrEP helps these people to stay together since the negative person is sure that he/she is protected. This negative person provides support to the positive person to continue adhering well since the positive person has an aim of suppressing the virus. This HIV positive person then adheres well to treatment such that the virus is suppressed, and also to maintain the relationship with the partner.*

-Female counselor, Age 34

*When he tested negative and I tested positive he agreed to take PrEP to encourage me to take ART. He takes PrEP because of me so I am supposed to take ART to protect him. That encouragement he gives me motivates me to take ART. . . Him taking PrEP every day at the same time with me even though he is negative motivates me so much to take ART.*

-Female partner living with HIV, Age 29

**Reluctance to embrace PrEP.** Despite efforts to engage couples in integrated services, providers openly acknowledged the reluctance of some HIV-negative partners to initiate PrEP. Some did not believe in the possibility of HIV serodifference or assumed HIV acquisition was inevitable. Others decided PrEP was not needed because they were still HIV-negative after years of being with a partner living with HIV. In "newer" relationships, serodifference upended commitments to the relationship, with couples choosing to separate rather than take PrEP. Also prominent were concerns embedded in taking pills for HIV prevention. Some partners simply never quite "bought in" to the efficacy of PrEP. Others complained about pill burden, or feared they would be unable to take daily tablets consistently. Many were worried they would be misidentified as someone living with HIV, due to the similarity in appearance between PrEP and ART.

*. . .As you know, HIV is associated with some stigma. They think if people see them with the tin (bottle of PrEP), they will think that they are HIV positive so they may discriminate them due to that. The packaging, the daily dosage and the pills are just like ART.*

-Female counselor, Age 38

For these same reasons, it was common for HIV-negative partners to initiate PrEP, only to drop out of care before collecting refills. In this qualitative sample, some individuals continued PrEP for several months, only to stop after their initial interest waned, or they decided PrEP protection was no longer needed–e.g., when separating from a partner or spending a prolonged period of time away from home. While PrEP discontinuation was expected when partners were no longer together, providers reported that retention on PrEP was one of the greatest "ongoing challenges" they faced in implementing integrated delivery.

## Discussion

We evaluated implementation of an integrated strategy of ART and PrEP delivery for serodifferent couples in 12 public health ART clinics in and around Kampala, Uganda. Using qualitative methods, we identified processes contributing to program implementation, from the points of view of implementing health care providers and the couples who benefitted from the integrated delivery strategy. High quality training, technical supervision, and teamwork supported the viability of integrated delivery, while increased demands on staff interfered with PrEP implementation. Interest in the PrEP program was promoted through the numerous ways health care providers made integrated ART and PrEP meaningful for serodifferent couples. Tailored counseling messages, efforts to build confidence in integrated delivery, and strategies to create demand for PrEP strengthened couples' enthusiasm. Couples in the qualitative sample responded positively to providers' efforts; HIV-negative partners initiated PrEP to preserve their relationships, inspiring continued commitments to ART adherence among their spouses. Lack of disclosure among couples and poor retention on PrEP were identified as implementation barriers

Qualitative investigations are valuable for examining implementation processes [21–23]. From the perspectives of implementing providers, our qualitative study unpacked specific processes critical to integrating PrEP delivery into ART services, offering insights into intervention feasibility. In interviews, providers emphasized the value of ongoing training and TA from the Partners PrEP Program team, which built competence and confidence in PrEP delivery and created a supportive environment that was highly motivating. Providers also spoke about the importance of teamwork in overcoming human resource challenges and managing the extra time PrEP provision required. This collaborative approach inspired a sense of commitment and "ownership" that was pivotal to feasibility. The PrEP program also had high levels of engagement at the organizational level, and from local community and government stakeholders [17]. Strong partnerships among stakeholders has been identified as contributing to implementation success in other PrEP delivery contexts [24].

Few other in-depth examinations of implementation processes of integrated ART and PrEP delivery have been described in the literature. Most related to the present work is an analysis of TA report data collected during routine supervision visits at Partners PrEP Program facilities [25]. Several similar process-related constructs were identified in that analysis, including supportive supervision during TA visits, recognition of facility "champions," opportunities for facility staff to discuss the implementation process and proactive strategies to engage intervention participants. In Kenya, where PrEP was scaled out to 25 public health clinics nationally, qualitative data were used in combination with TA reports to characterize the process of PrEP service integration [26]. The team found that ART facilities made numerous adaptations to simplify and strengthen PrEP delivery in response to clients' and providers' preferences and needs. Among these were "fast-tracked" services, on-the-job training for staff, and discussions of PrEP delivery in routine meetings. Our findings provide a complementary perspective on the importance of training, collaboration, and engagement, and extend previous work by accentuating the significance of these processes for feasibility.

The Partners PrEP Program was designed specifically for serodifferent couples, with their needs and priorities in mind. In promoting integrated delivery as a resolution to the "discordance dilemma" [27], providers demonstrated to couples they were attuned to the importance of relationship dynamics and were invested in limiting separations. This resonated with couples and made them feel valued and heard. "Person-centered" approaches to services are recognized as an essential component of high-quality healthcare [28–30], particularly in HIV prevention and treatment [31–33]. Person-centered services respect individual preferences,

are contextually-appropriate, and place wellbeing, agency and empowerment at the center of the health response. Our data shed light on ways implementing health care providers embraced this concept and tailored integrated ART and PrEP delivery to make it more meaningful to serodifferent couples.

By recognizing the meaning of PrEP for relationships, providers helped to promote acceptability of integrated delivery for serodifferent couples. Frameworks for understanding acceptability in HIV prevention and treatment services are varied and have evolved over time [34]. Earlier assessments of acceptability have focused on product attributes and characteristics [35, 36], ease of or intention to use an intervention [37], and behavior change [38]. The Theoretical Framework of Acceptability (TFA) was developed to guide the assessment of acceptability, representing it as a "multi-faceted construct that reveals the extent" an intervention is appropriate from the perspective of implementers or its recipients [39]. Our qualitative study is the first that we know of that calls attention to the significance of "meaning" as a component of acceptability. For serodifferent couples in the qualitative sample, the integrated delivery strategy was acceptable because it was meaningful to them in the context of their priorities and values. This is aligned with the concept of "ethicality" in the TFA, which considers the fit of the intervention within individuals' value systems [39].

Efforts to assess implementation of evidence-based interventions have historically been limited by the lack of clear, consistent terminology or systematic description of evaluation approaches [40–42]. To address this gap, Proctor and colleagues proposed a framework of eight "outcomes" to evaluate implementation process: acceptability, adoption, appropriateness, feasibility, fidelity, cost, penetration and sustainability [43]. In our analysis, feasibility and acceptability emerged as most salient for understanding processes of integrating PrEP into ART services. Our data also highlight the multidimensionality of these implementation concepts in practice. For example, features of the PrEP program that made it feasible also contributed to adoption by health care providers, potentially facilitating sustainability within public health facilities. Acceptability was bolstered by providers' commitments to improving the "fit" of the integrated delivery strategy and make it more appealing to serodifferent couples. The overlap we observed suggests that implementation outcomes are interrelated in complex ways and may not be conceptually distinct.

One objective of the PrEP program was to investigate whether PrEP use could be understood as a "modeled behavior." Behavior modeling refers to the act of observing a role model perform a behavior, and using this information to guide subsequent behaviors. The qualitative data reported here inform the dynamics of pill-taking in couples by revealing the significance of PrEP initiation for adherence to ART. HIV-negative partners accepted PrEP to signify their continuing commitment to the partnered relationship. Inspired by this gesture, and relieved not to be facing separation, partners living with HIV reciprocated by recommitting themselves to ART adherence. This same dynamic, in which serodifferent partners used antiretroviral medication to mutually signal commitment, was evident in our previous research on ART and PrEP use by Ugandan serodifferent couples [13], and may help to explain the high levels of PrEP initiation and sustained ART use observed in the stepped-wedge cluster randomized trial [17].

A major strength of this analysis is its focus on the perspectives of program implementers and serodifferent couples. This detailed examination of implementation processes has revealed how integrating PrEP into existing ART services was feasible for providers. It also highlights the efforts they made to promote acceptability for couples, including a new emphasis on the "meaning" of the intervention to recipients. This study has the following limitations. The Partners PrEP Program delivered oral PrEP to HIV-negative members of serodifferent couples in urban Uganda. Study findings may not apply to other priority populations in Uganda or other

delivery approaches with newer PrEP modalities. Although data were collected throughout program implementation, interviews with health care providers were conducted near study completion and observations about earlier stages of implementation may not have been captured in this analysis. Finally, we acknowledge the possibility of social desirability bias. Some interview participants may have intentionally characterized their responses to the integrated strategy and/or ART and PrEP use in a positive light.

## Conclusion

As the HIV prevention landscape continues to evolve and new PrEP modalities and delivery approaches come to scale, it becomes critical to understand what makes these interventions work or not work. This analysis highlights the processes involved in implementing integrated ART and PrEP delivery for serodifferent couples in central Ugandan public health clinics. It shows in detail what made it feasible for health care providers to implement this prevention approach, and foregrounds how they adapted the strategy to make it acceptable to couples. Framing ART and PrEP use as a way of addressing couples' personal priorities made the integrated strategy meaningful in the local context, and was especially noteworthy. A greater emphasis on understanding the meaning of PrEP for users and its contribution to implementation promises to strengthen future research on PrEP scale up in sub-Saharan Africa.

## Supporting information

**S1 Checklist. Inclusivity in global research checklist.**
(PDF)

**S1 File. The partners PrEP program study protocol.** The qualitative study was part of behavioral research collected in the stepped-wedge cluster randomized trial.
(PDF)

**S2 File. Interview guide—Initial interview, index participant.** This qualitative interview guide was used for initial interviews with partners living with HIV who were taking ART.
(PDF)

**S3 File. Interview guide—Initial interview, partner participant.** This interview guide was used for initial interviews with HIV-negative partners who initiated PrEP.
(PDF)

**S4 File. Interview guide—Follow-up interview, index and partner participants.** This interview guide was used for follow-up interviews with a subsample of participants living with HIV and their HIV-negative partners.
(PDF)

**S5 File. Interview guide—Health care provider participants.** This interview guide was used for interviews with health care providers from public health facilities implementing integrated ART and PrEP delivery.
(PDF)

**S6 File. Codebook.** This codebook was used to guide the coding process. Codes were derived inductively from interview data.
(PDF)

**S7 File. Standards for Reporting Qualitative Research (SRQR) checklist.**
(PDF)

## Acknowledgments

We are grateful to the couples and health care providers who participated in qualitative interviews and shared their experiences and time with us. We also thank the contributions of the Partners PrEP Program Study Team: (University of Washington, Seattle, USA): Renee Heffron (protocol chair), Jared M. Baeten, Jane Simoni, Deborah Donnell, Ruanne Barnabas, Katherine K. Thomas, Dorothy Thomas, Erika Feutz, Cole Grabow, Allison Meisner, Kristin Ciccarelli, Caitlin Scoville, Katrina Ortblad; (Infectious Diseases Institute, Kampala, Uganda):

Andrew Mujugira, Timothy R. Muwonge, Joseph Kibuuka, Lylianne Nakabugo, Florence Nambi, Mai Nakitende, Diego Izizinga, Vicent Kasiita, Brenda Kamusiime, Alisaati Nalumansi, Collins Twesige, Grace K. Nalukwago, Charles Brown, Sylvia Namanda; (Uganda Ministry of Health, Kampala, Uganda): Herbert Kadama; (Harvard Medical School, Boston, USA): Norma C. Ware, Monique A. Wyatt, Emily Pisarski; (Brigham & Women's Hospital, Boston, USA): Ingrid T. Katz.

## Author Contributions

**Conceptualization:** Monique A. Wyatt, Andrew Mujugira, Renee Heffron, Norma C. Ware.

**Data curation:** Monique A. Wyatt, Emily E. Pisarski, Alisaati Nalumansi, Vicent Kasiita, Brenda Kamusiime, Grace K. Nalukwago.

**Formal analysis:** Monique A. Wyatt, Emily E. Pisarski.

**Funding acquisition:** Renee Heffron.

**Investigation:** Alisaati Nalumansi, Vicent Kasiita, Brenda Kamusiime, Grace K. Nalukwago.

**Methodology:** Monique A. Wyatt, Norma C. Ware.

**Project administration:** Monique A. Wyatt, Emily E. Pisarski, Dorothy Thomas, Timothy R. Muwonge.

**Supervision:** Monique A. Wyatt, Emily E. Pisarski, Andrew Mujugira, Norma C. Ware.

**Writing – original draft:** Monique A. Wyatt.

**Writing – review & editing:** Monique A. Wyatt, Emily E. Pisarski, Andrew Mujugira, Renee Heffron, Norma C. Ware.

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
