## [Decision Letter · Decision Letter 0]

10 Oct 2023

PGPH-D-23-01404

How PrEP delivery was integrated into public ART clinics in central Uganda: A qualitative analysis of implementation processes

Dear Dr. Wyatt,

Thank you for submitting your manuscript to PLOS Global Public Health. After careful consideration, we feel that it has merit but does not fully meet PLOS Global Public Health’s publication criteria as it currently stands. Therefore, we invite you to submit a revised version of the manuscript that addresses the points raised during the review process.

We look forward to receiving your revised manuscript.

Kind regards,

Sarah E. Brewer, PhD

Academic Editor

Journal Requirements:

Additional Editor Comments (if provided):

Reviewers' comments:

Reviewer's Responses to Questions

**Comments to the Author**

1. Does this manuscript meet PLOS Global Public Health’s publication criteria? Is the manuscript technically sound, and do the data support the conclusions? The manuscript must describe methodologically and ethically rigorous research with conclusions that are appropriately drawn based on the data presented.

Reviewer #1: Yes

Reviewer #2: Yes

2. Has the statistical analysis been performed appropriately and rigorously?

Reviewer #1: Yes

Reviewer #2: N/A

3. Have the authors made all data underlying the findings in their manuscript fully available (please refer to the Data Availability Statement at the start of the manuscript PDF file)?

Reviewer #1: Yes

Reviewer #2: No

4. Is the manuscript presented in an intelligible fashion and written in standard English?

Reviewer #1: Yes

Reviewer #2: Yes

5. Review Comments to the Author

Reviewer #1: Thank you for the opportunity to review this important manuscript.

This is a qualitative study near Kampala, Uganda. The study was nested within a stepped-wedged cluster randomized trial of the integration of PrEP and ART delivery for sero different couples from the perspectives (in-depth interviews) of 42/149 couples (83 interviews) and 36 healthcare providers from 12 health facilities. A second interview was conducted in a subset of 23 couples (45 interviews).

Negative partners-initiated PrEP to support and preserve relationships, encouraging positive partners to adhere to ART.

Introduction

1. Line 63 – “near elimination of HIV”, not clear of the meaning, can be clarified by the authors to improve comprehension?

Analyses and findings.

1. Is there data on education and occupation available for Sero different couples’ members?

2. Is there data on health professionals' cadres?

3. Stigma is well described as one of the barriers to PrEP, any mention from the couples about fear of Adverse effects?

4. Was data saturation discussed/achieved?

Reviewer #2: Thank you for inviting me to review this manuscript. The article meets the main criteria for publication as articulated by the journal, in that, for example, it describes original research, is clearly written in English, and the study went through ethical review / an IRB etc. The qualitative sample is a robust size and is derived from a well designed stepped wedge cluster RCT and the description of the analyses are appropriate and detailed. I enjoyed reading the discussion, and feel that the conclusions are supported by the data.

That said, there are a few main issues I would like to raise:

(1) This is a purposively selected sample and as such the data are not necessarily representative of the larger community of serodifferent couples (which the authors acknowledge). However, I would still like to know more about how the participants were selected (as well as a sense of response rates to the invitations to participate).

(2) The authors indicate that in fact many of the couples did not return for PrEP / continue on PrEP. This is a major challenge for the field in general, and should be highlighted more in the reporting of results and implications related to the current findings. If issues related to maintenance of relationships are key for PrEP initiation (as is reported here), would they not be for PrEP continuation / persistence? Why would this be the case? Further, the authors state at one point that providers reported that a lot of people resisted initiating (and not just remaining on) PrEP. How should that information be interpreted / what are the implications for the other PrEP initiation findings highlighted by the authors?

(3) This manuscript seems overly long to me, and I feel like the main findings that bring nuance / a new perspective (and hence the main contributions) are getting somewhat lost. Many of the points / themes raised seem pretty generic (or known) to me – for example, the importance of high quality training or teamwork, or tailored counseling messages. The finding that is most interesting to me relates to how participants often perceived that initiating PrEP could / would preserve their relationships and that many partners with HIV recommitted to ART because the partner not living with HIV was willing to stay in the relationship and use PrEP. I would consider shortening the manuscript substantially, to highlight the important results.

(4) Data are not really available to the public – they need to be requested and will ‘be considered on a case by case basis.’ While this is not uncommon for qualitative research, it is important to be aware that this is the case, given related requirements of the journal.

The following additional comments are categorized by manuscript section.

Abstract: My sense is that the abstract doesn’t adequately highlight the most interesting / unique results. In addition, the results concerning relationship dynamics are not really explained clearly.

Introduction: The text states that the intervention was provided in public health facilities under ‘real world conditions.’ While the intervention was implemented in public facilities it seems that the activities were implemented by specially trained staff supported over time by the PrEP study team. If that is correct, then one could argue that this is not really ‘real world’ conditions – please do clarify / nuance the related text.

Methods: The text states that transcripts were reviewed for quality. Can the authors clarify who reviewed them, how many people reviewed the transcripts, and how they were reviewed?

It would be useful to have some sort of conceptual framework, a guide for the types of issues that were explored (in the context of this Implementation Science study / process evaluation.) Further, the text refers to a ‘framework method’ to ‘visually display coded data’ – could the authors explain this a bit?

Results: Given that only 10% of initiators continued PrEP at 6 months, the text should very much highlight that these results are only abut PrEP initiation and not PrEP ‘use’ more broadly or continuation.

Tables 1 and 2 aren’t so useful, to my mind, as they are reporting basic percentages that are also described in the text.

Regarding the theme of providers being concerned about additional work / burden, this is a real and commonly reported finding, but I am not quite clear how it is it relevant for the main question on how to integrate PrEP into ART programming. Can the authors clarify?

Discussion: The summary in the beginning of the discussion is quite generic regarding provider feedback – the contribution of these findings is not quite clear to me.

6. PLOS authors have the option to publish the peer review history of their article (what does this mean?). If published, this will include your full peer review and any attached files.

**Do you want your identity to be public for this peer review?** For information about this choice, including consent withdrawal, please see our Privacy Policy.

Reviewer #1: No

Reviewer #2: No

---

## [Editor Report · Decision Letter 1]

22 Jan 2024

How PrEP delivery was integrated into public ART clinics in central Uganda: A qualitative analysis of implementation processes

PGPH-D-23-01404R1

Dear Ms. Wyatt,

We are pleased to inform you that your manuscript 'How PrEP delivery was integrated into public ART clinics in central Uganda: A qualitative analysis of implementation processes' has been provisionally accepted for publication in PLOS Global Public Health.

Best regards,

Sharmistha Mishra, M.D., Ph.D

Academic Editor